# Cerebral Protection in TAVR—Can We Do Without? A Real-World All-Comer Intention-to-Treat Study—Impact on Stroke Rate, Length of Hospital Stay, and Twelve-Month Mortality

**DOI:** 10.3390/jpm12020320

**Published:** 2022-02-21

**Authors:** Carolina Donà, Matthias Koschutnik, Christian Nitsche, Max-Paul Winter, Veronika Seidl, Jolanta Siller-Matula, Markus Mach, Martin Andreas, Philipp Bartko, Andreas Anselm Kammerlander, Georg Goliasch, Irene Lang, Christian Hengstenberg, Julia Mascherbauer

**Affiliations:** 1Department of Internal Medicine II, Division of Cardiology, Medical University of Vienna, 1090 Vienna, Austria; carolina.dona@meduniwien.ac.at (C.D.); matthias.koschutnik@meduniwien.ac.at (M.K.); christian.nitsche@meduniwien.ac.at (C.N.); max-paul.winter@meduniwien.ac.at (M.-P.W.); veronika.seidl@meduniwien.ac.at (V.S.); jolanta.siller-matula@meduniwien.ac.at (J.S.-M.); philipp.bartko@meduniwien.ac.at (P.B.); andreas.kammerlander@meduniwien.ac.at (A.A.K.); georg.goliasch@meduniwien.ac.at (G.G.); irene.lang@meduniwien.ac.at (I.L.); christian.hengstenberg@meduniwien.ac.at (C.H.); 2Department of Cardiac Surgery, Medical University of Vienna, 1090 Vienna, Austria; markus.mach@meduniwien.ac.at (M.M.); martin.andreas@meduniwien.ac.at (M.A.); 3Cardiovascular Imaging Research Center, Massachusetts General Hospital and Harvard Medical School, Boston, MA 02115, USA; 4Department of Internal Medicine 3, University Hospital St. Poelten, Karl Landsteiner University of Health Sciences, 3500 Krems, Austria

**Keywords:** Sentinel™ Cerebral Protection System, transcatheter aortic valve replacement, stroke, mortality

## Abstract

**Background**: Stroke associated with transcatheter aortic valve replacement (TAVR) is a potentially devastating complication. Until recently, the Sentinel™ Cerebral Protection System (CPS; Boston Scientific, Marlborough, MA, USA) has been the only commercially available device for mechanical prevention of TAVR-related stroke. However, its effectiveness is still undetermined. **Objectives:** To explore the impact of Sentinel™ on stroke rate, length of hospital stay (LOS), and twelve-month mortality in a single-center, real-world, all-comers TAVR cohort. **Material and Methods:** Between January 2019 and August 2020 consecutive patients were assigned to TAVR with or without Sentinel™ in a 1:1 fashion according to the treating operator. We defined as primary endpoint clinically detectable cerebrovascular events within 72 h after TAVR and as secondary endpoints LOS and 12-month mortality. Logistic and linear regression analyses were used to assess associations of Sentinel™ use with endpoints. **Results:** Of 411 patients (80 ± 7 y/o, 47.4% female, EuroSCORE II 6.3 ± 5.9%), Sentinel™ was used in 213 (51.8%), with both filters correctly deployed in 189 (46.0%). Twenty (4.9%) cerebrovascular events were recorded, ten (2.4%) of which were disabling strokes. Patients with Sentinel™ suffered 71% less (univariate analysis; OR 0.29, 95%CI 0.11–0.82; *p* = 0.02) and, respectively, 76% less (multivariate analysis; OR 0.24, 95%CI 0.08–0.76; *p* = 0.02) cerebrovascular events compared to patients without Sentinel™. Sentinel™ use was also significantly associated with shorter LOS (Regression coefficient −2.47, 95%CI −4.08, −0.87; *p* < 0.01) and lower 12-month all-cause mortality (OR 0.45; 95%CI 0.22–0.93; *p* = 0.03). **Conclusion:** In the present prospective all-comers TAVR cohort, patients with Sentinel™ use showed (1) lower rates of cerebrovascular events, (2) shortened LOS, and (3) improved 12-month survival. These data promote the use of a CPS when implanting TAVR valves.

## 1. Background

Almost 4% of individuals over 70 and more than 9% over 80 years of age suffer from severe aortic stenosis (AS) [1]. Transcatheter aortic valve replacement (TAVR) has revolutionized treatment options in this common disease, especially in patients with high surgical risk. The technology has become widely applicable and safe through the availability of various self-expanding and balloon-expandable devices designed to fit almost every anatomy [2]. Recent guideline recommendations have greatly expanded the indication for TAVR, particularly in low-risk and younger age groups [3]. Despite ongoing improvements of the TAVR procedure, significant risks remain. One of the most dangerous complications is stroke, which has been reported to affect up to one in ten patients [4,5,6,7,8]. Factors that have been shown to predispose for TAVR-associated cerebrovascular events include age, atrial fibrillation, history of cerebrovascular events, very severe and calcified AS, and implantation of more than one valve [4,5,7]. However, no widely accepted algorithms currently exist that would allow the prediction of TAVR-related cerebrovascular events. To protect the supra-aortal arteries from embolizing material during the procedure, mechanical cerebral protection systems (CPS) have been developed. Until recently, only the Sentinel™ (Boston Scientific) CPS has been commercially available. However, data on the effectiveness of Sentinel™ are conflicting. While several small studies failed to demonstrate a reduction of clinically relevant stroke in TAVR with Sentinel™ [9,10,11], three larger studies reported significant reductions in cerebrovascular event [12,13,14] and also in-hospital mortality [11,12] rates. The reasons for these discrepancies as well as the limitations of the Sentinel™ CPS have so far not been carefully investigated. In particular, it remains unclear whether Sentinel™ also impacts the length of hospital stay (LOS) and twelve-month mortality. Thus, the effectiveness of Sentinel™ is still undetermined, and it has, as a result, not evolved as standard-of-care. 

In this prospective study, we investigated the efficacy of Sentinel™ using clinically detectable cerebrovascular events within 72 h after TAVR as primary and LOS and 12-month mortality as secondary endpoints.

## 2. Methods

### 2.1. Study Population

We enrolled consecutive patients with AS scheduled for transfemoral TAVR at the General Hospital Vienna, a university-affiliated tertiary center. The multi-disciplinary heart team confirmed the eligibility and the decision for TAVR according to current guidelines [15]. Patients were assigned to TAVR with or without Sentinel™ according to the first operator as two out of four first operators systematically used the device whenever possible, while the other operators did not. Procedural complications and in-hospital clinical outcomes were assessed according to the Valve Academic Research Consortium-2 (VARC-2) criteria [16]. The patients were followed for at least 12 months. The study was approved by the local ethics committee (EK No. 2218/2016). All patients provided written informed consent. 

### 2.2. Transcatheter Aortic Valve Replacement 

Transfemoral TAVR procedures were performed according to institutional standards in local or general anesthesia in a hybrid catheterization laboratory. According to current guideline recommendations, oral anticoagulation was discontinued before the procedure. During TAVR, patients received unfractionated heparin with a target activated clotting time of 250 s. Antiplatelet therapy, if present, was continued. The percentages of antiplatelet therapy used refer to the preprocedural therapy, postinterventionally, DAPT was started according to 2017 ESC Guidelines for the Management of Valvular Heart Disease [15].

### 2.3. Sentinel™ Cerebral Protection Device

The Sentinel™ device (Boston Scientific^©^) is a dual-filter-based intra-luminal embolic protection device inserted through a 6-French sheath via the right radial, ulnar or brachial artery. The proximal filter has to be positioned in the brachiocephalic trunk and the distal filter in the left common carotid artery. Both filters consist of a radiopaque nitinol frame with a 140 µm pore polyurethane filter. The two filters cover more than 90% of the cerebral blood flow, excluding only the left vertebral artery’s territory [17]. One study reported that 74% of the brain volume is totally, 24% is partially protected, leaving only 2% of brain volume unprotected [18]. Prior to the procedure, brachiocephalic trunk and carotid artery anatomy were analyzed on computed tomography angiography. Sentinel™ use was not attempted in the presence of severe left carotid artery stenosis, defined as stenosis severity >70%, or if it was known that right radial access was not possible. According to the protocol, ulnar, brachial, and axillar access was not used. The Sentinel™ system was positioned after insertion of the TAVR sheath and removed after valve deployment, prior to femoral closure. 

### 2.4. Histopathology

Immediately after the procedure, debris from the Sentinel™ filters were collected, digitally photographed and stored in 7.5% neutral buffered formalin. Particles with a diameter >2 mm were dehydrated and embedded in paraffin. The paraffin block was cut into 3µm slices, which were stained with modified trichrome [19].

### 2.5. Definition of Endpoints

Cerebrovascular events were classified according to VARC-2 criteria, including disabling and non-disabling stroke as well as transitory ischemic attack [16]. The primary endpoint was defined as any clinically detectable cerebrovascular event within 72 h of TAVR. [13,20] During these 72 h, patients were evaluated clinically every 6 to 12 h [4,12,21,22] by a board-certified cardiologist. Face palsy, arm weakness, speech changes, eye deviation or denial were assessed according to the FAST-ED scale [23]. In case of any sign of neurological compromise, a comprehensive neurological assessment, including the modified Rankin scale as well as the National Institute of Health Stroke Scale (NIHSS), was performed by a board-certified neurologist, similar to the approach used by Seeger et al. [12]. This assessment also included cerebral computed tomography and/or cerebral magnetic resonance imaging (cMRI), interpreted by board-certified neuroradiologists. The final diagnosis of a cerebrovascular event was adjudicated by a board-certified neurologist. All-cause mortality at 12 months and LOS in days were chosen as secondary endpoints. 

LOS was defined as overall length of hospital stay post-procedure, including the stay at the tertiary care center and any immediate subsequent hospital stay. After discharge, patients were followed in our outpatient clinic at three and twelve months. 

### 2.6. Statistical Analysis 

Categorical parameters are presented as counts and percentages, continuous variables as mean ± standard deviation. To compare baseline variables between groups, χ^2^ test or Fisher’s exact test and Wilcoxon ranks sum test were applied as appropriate. 

For the primary endpoint, we used logistic regression models to identify baseline variables associated with cerebrovascular events. For the assessment of differences in LOS and 12-month mortality, linear as well as univariate and multivariate Cox regression models were applied. In addition, we used Kaplan–Meier estimates and the log-rank test to report differences in all-cause mortality. Variables with a *p*-Value < 0.05 were entered in the multivariate analysis using a simultaneous approach.

In a separate step, we performed a propensity score analysis using the psmatch2 command in Stata 15.1 (StataCorp., College Station, TX, USA) using EuroSCORE II, atrial fibrillation (AF), diabetes mellitus, gender, coronary artery disease (CAD) and valve type (self-expanding versus balloon-expandable), similar to that previously used by Seeger et al. [20]. A *p*-Value < 0.05 was considered statistically significant, and tests were two-sided. Statistical analysis was performed using SPSS 26.0. 

## 3. Results

### 3.1. Patient Population 

Between January 2019 and August 2020, 411 consecutive TAVR patients were enrolled. The mean age was 80.4 ± 6.7 years, 47.4% were female. A prior history of cerebrovascular events was present in 7.4% of patients. 

Figure 1 and Panel A of Figure 2 depict patient flow. Out of our 411 patients, Sentinel™ was used in 213 (51.8%, CPS). Of these, the device was correctly deployed in 189 individuals (46.0% in total; 88.7% of CPS; CPS+). In 24 patients (5.8% of the total study cohort; 11.3% of CPS; CPS−) only the proximal filter could be positioned correctly in the brachiocephalic trunk while the distal filter remained closed in the ascending aorta. 198 (48.2%) underwent TAVR without CPS (noCPS). 

Table 1 displays baseline characteristics, which were well balanced across groups (noCPS vs. CPS). Differences were only observed with regard to mean and peak aortic valve gradients. No differences were found regarding procedural parameters except for fluoroscopy time, which was longest in CPS− patients. No local vascular complications were attributed to CPS use. 

### 3.2. Cerebrovascular Events

In total, 20 (4.9%) cerebrovascular events occurred, of which 10 (2.4%) were disabling strokes with a modified Rankin Scale >2 at 90 days (Table 2). In the Sentinel™ group cerebrovascular events were reduced by 70% (noCPS 7.6% vs. CPS 2.3%, *p* = 0.02) (Figure 2, Panel C). 

In the Sentinel™ group (CPS) five cerebrovascular events were recorded. Of these, two occurred in patients with correctly positioned CPS (CPS+; 2/189, 1.1%). Both comprised cerebral regions supplied by the left vertebral artery. One stroke was disabling with a modified Rankin Scale > 2 at 90 days. In the CPS− group, three events occurred (3/24; 12.5%). In two of them, brain areas supplied by the left carotid or vertebral artery were affected, which were not covered by the distal filter. In one, the event was defined as disabling. In the third patient, who also suffered a disabling stroke, the right mid cerebral artery was occluded, indicating incomplete coverage of the right brachiocephalic trunk/the right carotid artery. 

In the multivariate analysis, Sentinel was significantly associated with a decreased stroke rate after 72 h (OR 0.24, 95%CI 0.08–0.76; *p* = 0.02), whereas implantation of more than one valve was the only factor significantly associated with a higher cerebrovascular event rate (OR 16.7, 95%CI 2.69–103.92; *p* < 0.01; Table 3).

On propensity score-matched analysis, Sentinel™ remained significantly associated with lower cerebrovascular event rate (OR 0.12, 95%CI 0.03–0.51; *p* < 0.01, Appendix A). When the analysis was restricted to patients who underwent TAVR with Sentinel™, correct CPS placement remained the only parameter independently associated with cerebrovascular events (OR 0.06, 95%CI 0.01–0.43; *p* < 0.01).

### 3.3. Timing of Stroke

Of 20 cerebrovascular events, 16 (80%) events were considered acute (within 24 h of TAVR) [5,24,25]. In this group, no CPS+ patients suffered an event. Four strokes occurred 25 to 72 h after TAVR. (Figure 3).

Both patients with a cerebrovascular event in the CPS+ group suffered a sub-acute stroke (at 30 and 48 h after TAVR, respectively). The three events in the CPS− group occurred at 0, 19, and 27 h.

All other events were observed in patients without CPS.

### 3.4. All-Cause Mortality at 12 Months

At 12 months, 51 patients (12.4%) had died. Sentinel™ use was associated with a 45% reduction of all-cause mortality at twelve months (noCPS 16.2% vs. CPS 8.9%, *p* = 0.026). In patients with incorrectly/incompletely positioned Sentinel™ (CPS−), mortality at 12 months was similar to TAVR without CPS (16.7%, Figure 2, Panel B; Table 2). 

In the univariate Cox regression analysis, Sentinel™ was significantly associated with a reduced twelve-month mortality (OR 0.48, 95%CI 0.26–0.87; *p* = 0.02). In the multivariate analysis, Sentinel™ remained significantly associated with one-year mortality (OR 0.45, 95%CI 0.22–0.93, *p* = 0.03) as well as procedure time (OR 1.02, 95%CI 1.01–1.03; *p* < 0.01; Table 4).

In the propensity score-matched analysis, Sentinel™ remained significantly associated (OR 0.48, 95%CI 0.27–0.88; *p* = 0.02; Appendix A) with all-cause mortality at 12 months. 

### 3.5. Length of Hospital Stay

After TAVR, patients stayed in hospital for 7.5 ± 8.0 days. Overall, the systematic use of Sentinel™ decreased the LOS after TAVR by 20% (noCPS 8.4 ± 9.6 vs. CPS 6.7 ± 6.1 days; *p* = 0.03; Figure 2, Panel D). Incorrect/incomplete CPS deployment did not change the LOS (noCPS 8.4 ± 9.6 vs. CPS− 8.6 ± 9.9 days; *p* = 0.92; Table 5).

Patients with cerebrovascular events required significantly longer in-hospital stays than patients without (no event 7.1 ± 6.7 vs. event 16.4 ± 19.1 days; *p* < 0.01). In addition, among patients who suffered cerebrovascular events those with Sentinel^TM^ required significantly shorter LOS (noCPS 7.7 ± 7.6 vs. CPS 6.4 ± 5.3 days; *p* = 0.05)

In the multivariate analysis, Sentinel^TM^ (Regression coefficient −2.47, 95%CI −4.08, −0.87; *p* < 0.01) as well as fluoroscopy time (Regression coefficient 0.20, 95%CI 0.10, 0.29; *p* < 0.01) were the only parameters significantly associated with LOS (Table 5). In the propensity score-matched analysis, Sentinel™ remained significantly associated (adj. Regression coefficient −2.005, 95%CI −3.561, −0.450; *p* = 0.01; Appendix A) with LOS.

### 3.6. Learning Curve

To assess a potential impact of a learning curve for the use of Sentinel™, patients were divided into two groups comprising the first and the second half of the study population. No differences with regard to rates of incorrect/incomplete CPS deployment were observed (1st half 11.2%, 2nd half 11.3%). Fluoroscopy time did also not significantly change from the first to the second half of patients (19 ± 9 min vs. 18 ± 9 min, *p* = 0.256). 

### 3.7. Histopathology

In 91% of filters, debris was captured. In 64%, particles were large enough (>2 mm) to allow comprehensive histopathological work-up. The most commonly isolated materials were valve tissue (62.5%) arterial wall (37.5%) and atherosclerotic plaque (32.5%). Rare findings included myocardium (10.0%), acute thrombus (7.5%), organized thrombus (7.5%) and foreign material (2.5%). Figure 4 shows typical examples of fresh debris, valve tissue, atherosclerotic plaque material, thrombus, and fibrous tissue. 

## 4. Discussion

In the present TAVR population we systematically evaluated the Sentinel™ CPS effectiveness and limitations and report three main findings: patients who underwent TAVR with Sentinel™ CPS showed (1) a 70% lower risk of stroke, (2) significantly shorter LOS, and (3) 42% lower all-cause mortality rates at twelve months. However, in 1 out of 10 patients, Sentinel™ deployment was incomplete/incorrect. In these patients, cerebrovascular event, LOS, and mortality rates were as high/long as in patients receiving TAVR without Sentinel™.

The rapidly growing use of TAVR for the treatment of aortic valve disease, including young and low-risk populations [3], increases the necessity to make the procedure as safe as possible. One of the most devastating TAVR-related complications is stroke. Although rates of clinically overt cerebrovascular events associated with TAVR are reported to be low—particularly in low-risk patients [26]—they remain a major threat. Such events severely impair quality of life [27], increase mortality [4], and are associated with substantial healthcare costs [28]. Reported rates vary from 0.6% [14] to 10.0% [10], depending on patient population, study design, and stroke definition. As we treat more and more patients with TAVR and now seem to have a tool at hand that can potentially help to avoid cerebrovascular events in any age group, we are convinced that such tools should be used more broadly.

The Sentinel™ CPS has, until recently, been the only commercially available device for mechanical prevention of TAVR-related cerebrovascular events. However, conflicting data regarding its value and limitations have been reported [9,10,11,12,13,14]. Three small studies used cMRI to evaluate the impact of Sentinel™ on new brain lesions after TAVR [4,8,9]. In CLEAN-TAVI, such new lesions were reduced in the study arm [9], while in the SENTINEL and the MISTRAL-C trials no difference was found with regard to new lesion volume on cMRI with and without Sentinel™ [10,11]. Moreover, all three studies failed to show a reduction in clinically relevant cerebrovascular event rate with Sentinel™, although a strong trend was shown in the SENTINEL trial (5.6% vs. 9.1%, *p* = 0.25).

These studies were followed by three more recent larger trials [12,13,14]. In Sentinel-Ulm, including 802 patients, clinically overt stroke within 7 days after TAVR declined from 4.6% to 1.4% in the Sentinel^TM^ group (*p* = 0.03) [12], indicating a significant impact of CPS on cerebrovascular event rate. However, patients without CPS underwent TAVR during an earlier time span, thus representing a “historical cohort”, which causes potential bias (e.g. previous models of valves, experience of centers and operators). Similar results were reported from a large, but retrospective analysis of 1305 TAVR patients. Sentinel™ use was associated with a reduction of cerebrovascular events at 72 h of 65% (*p* < 0.01) and a reduction of the combined endpoint consisting of all-cause mortality and cerebrovascular events at 72 h of 66% (*p* < 0.01) [13]. In addition, a recent retrospective propensity-matched analysis by Megaly et al., comprising 1575 TAVR patients, showed a 76% reduction of cerebrovascular events with Sentinel^TM^ (*p* < 0.01). In-hospital mortality decreased from 1.0% to 0.0% (*p* = 0.04); however, LOS remained unchanged (*p* = 0.30) [14]. Conversely, Stachon et al. did not report a significant reduction of stroke rate in patients with Sentinel^TM^ (no Sentinel^TM^ 2.12% vs. Sentinel^TM^ 2.81%, *p* = 0.06) However, in that study patients with Sentinel^TM^ required significantly shorter LOS (13.87 ± 9.21 vs. 12.30 ± 7.54 days, *p* < 0.01) [29]. Both studies were retrospective observational analyses based on administrative data and included few patients that received Sentinel^TM^ (1.4% [14] and 3.8% [29] of the patient population, respectively). 

The LOS observed in our study (7.5 ± 8.0 days) is consistent with existing data reporting LOS in Europe. Kaier et al retrospectively examined 9345 patients and showed an average LOS of 17.5 days [30], similarly a recent study evaluating patients in the FRANCE-TAVI registry showed a median LOS of 7 (5–9) days [31]. Data from the National Inpatient Sample database showed a mean hospital stay of 5.7 ± 0.1days [32]. 

The present prospective real-world all-comers intention-to-treat study aimed to add to these previous reports by assessing (1) frequency and effect of incorrect/incomplete Sentinel™ deployment, (2) impact of systematic Sentinel™ use on LOS as well as (3) 12-month all-cause mortality. We showed here that Sentinel™ only offers protection if deployed correctly. A partial deployment was associated with event rates and duration of hospital stay similar to patients treated without CPS. Unfortunately, we were not able to define reliable predictors for successful placement, which was possible in 90% of individuals who were planned CPS-protected TAVRs. The small subgroup, in which correct Sentinel™ deployment was impossible (CPS−), seems to represent a high-risk population, presenting with particularly difficult anatomy due to atherosclerotic alterations such as severe kinking and/or calcification of the supra-aortic arteries. The assumption, that stroke in this subgroup may be caused by maneuvering of the CPS is possible, but, as SentinelÔ in general greatly reduced stroke rate, seems unlikely. However, the CPS− group was small (25 patients) therefore risk assessment is difficult. So far, incomplete deployment of Sentinel™ was only mentioned in the Sentinel™ trial; however, no association with stroke was shown [10].

The present study was the first to evaluate the effect of Sentinel™ CPS on twelve-month mortality, and clearly demonstrated a strong impact with a 45% reduction in all-cause death. It is still unclear whether clinically inapparent TAVR-associated cerebrovascular events may be more frequent than estimated and whether such events impact mid- and long-term mortality. However, our results bring previous TAVR studies on the effectiveness of Sentinel™ that used cMRI for the assessment of new brain lesions back into focus [4,9,10]. Although all of them failed to show an impact of Sentinel™ on clinically detected cerebrovascular events, the volume of new brain lesions on cMRI was reduced in CLEAN-TAVI [9]. In the two other studies, a trend towards a reduction in lesion volume with Sentinel™ was shown, which failed to reach statistical significance–potentially due to limited patient numbers (Sentinel trial: 121 treated with CPS [10]; MISTRAL-C: 32 treated with CPS [11]). Of note, several previous studies [10,11,33] reported the presence of debris in >90% of Sentinel™ filters, which is consistent with our findings. Thus, subclinical cerebral damage may be frequent during unprotected TAVR and may impair outcome. The impact of subclinical stroke on long-term outcome is unclear; however, several studies have shown, that silent embolism can lead to more pronounced cognitive decline as well as increased risk of dementia [34,35]. Therefore, an impact of CPS use on long-term outcome cannot be excluded. Following these results, we also showed here for the first time that the systematic use of Sentinel™ is associated with the duration of hospital stay after TAVR. 

Our study also showed limitations of the Sentinel™ device. Firstly, Sentinel™ does not provide complete protection of the entire brain, which limits its beneficial effect. Newly developed cerebral protection devices are currently entering the market, providing complete coverage of the cerebral vessels [36,37]. However, data on their effectiveness is more limited than for Sentinel™, and strokes despite use of these devices have been reported (2.2% to 6.4% of in-hospital stroke [36,37]). Further limitations refer to failed or incomplete Sentinel™ deployment in a significant proportion of our patients (11% deployment incomplete, 7% right radial/brachial access not possible). It furthermore cannot be out ruled that Sentinel™ placement and associated manipulations potentially cause cerebrovascular events.

## 5. Limitations

Several limitations merit comment. Given the single center nature, a selection bias cannot be excluded. However, advantages of single center studies include (a) adherence to a constant clinical routine, (b) constant quality of work-up, and (c) constant follow-up. Furthermore, patients were not randomized but pseudo-randomly assigned to TAVR with or without Sentinel™, irrespective of baseline characteristics. Cerebral MRI prior and after TAVR would have been the gold standard for the assessment of cerebral lesion volume. However, due to restricted cMRI capacity at our institution, we had to dispense systematic cMRI assessments. These were only performed in patients with clinical signs of cerebral compromise after TAVR. Finally, the small overall number of cerebrovascular events may limit generalizability of our findings, especially regarding patients with incompletely deployed Sentinel™.

## 6. Conclusions 

In the present prospective all-comers TAVR cohort, patients with systematic use of Sentinel™ showed (1) reduced rates of cerebrovascular events, (2) shortened LOS after TAVR, and (3) a reduction of all-cause mortality at 12 months. These data promote the use of a CPS when implanting TAVR valves. Long-term cost-effectiveness and the impact of anatomical constraints, necessitating customized procedural approaches, require further study.

### Impact on Daily Practice

**What is known:** The Sentinel™ Cerebral Protection System has been designed for mechanical prevention of transcatheter aortic valve replacement (TAVR)-related stroke. However, Sentinel™ limitations, as well as impact on twelve-month mortality and duration of hospital stay, are not well explored. 

**What is new:** In the present prospective all-comers cohort patients who underwent TAVR with Sentinel™ not only suffered fewer acute cerebrovascular events but also experienced shorter length of hospital stay and improved twelve-month survival. 

**What is next:** Long-term cost-effectiveness and the impact of anatomical constraints, requiring customized procedural approaches, require further study.

## Figures and Tables

**Figure 1 jpm-12-00320-f001:**
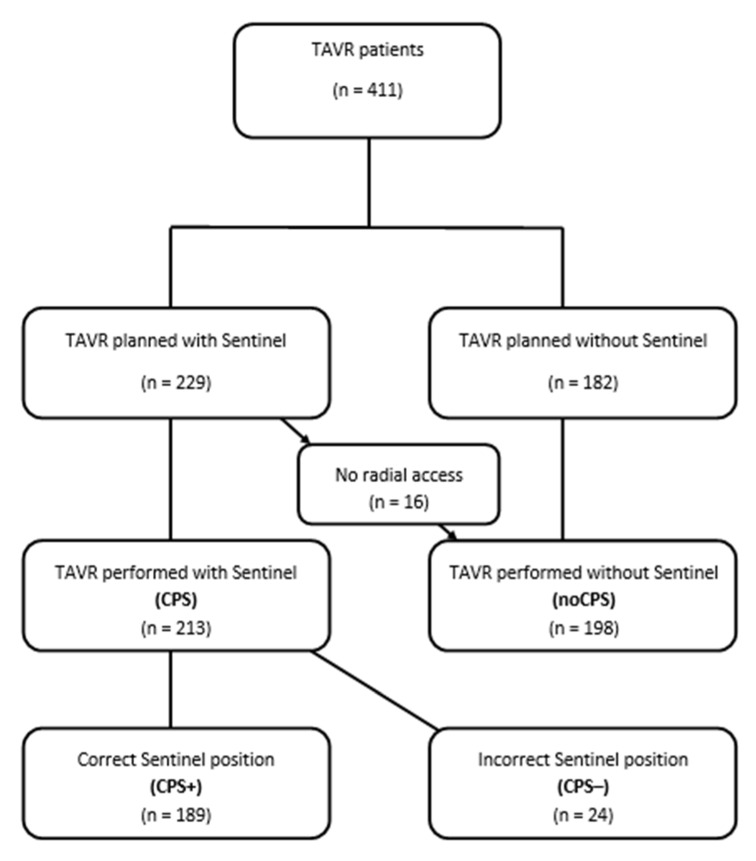
Patient flow. NoCPS indicates TAVR without Sentinel™, CPS indicates TAVR with Sentinel™. Patients in whom both filters were correctly deployed are presented as CPS+, patients with incomplete/incorrect CPS deployment are indicated as CPS−.

**Figure 2 jpm-12-00320-f002:**
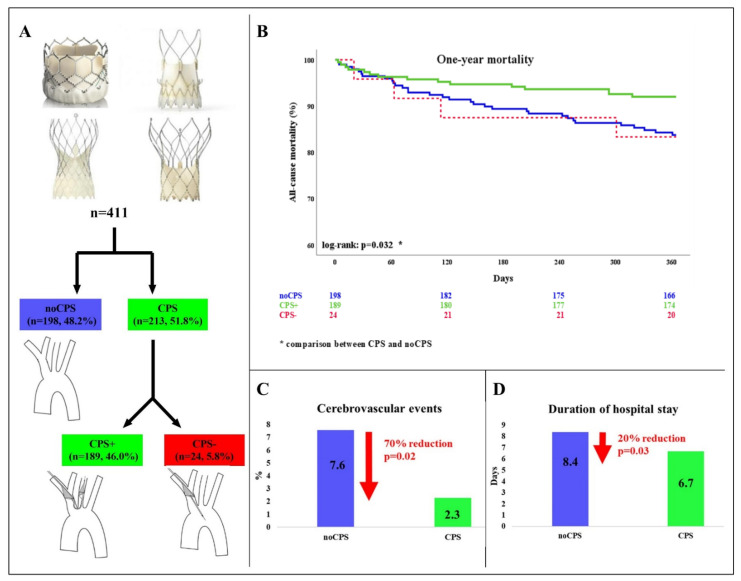
Study design and main results. NoCPS indicates TAVR without Sentinel™; CPS, TAVR with Sentinel™; CPS+, both filters deployed correctly; CPS–, incorrect/incomplete CPS deployment. Panel **A**, Patient flow; Panel **B**, Kaplan-Meier curves comparing twelve-month all-cause mortality of patients undergoing TAVR with and without Sentinel™; Panel **C**, Cerebrovascular event rate at 72 h after TAVR comparing noCPS and CPS; Panel **D**, Length of hospital stay comparing noCPS and CPS.

**Figure 3 jpm-12-00320-f003:**
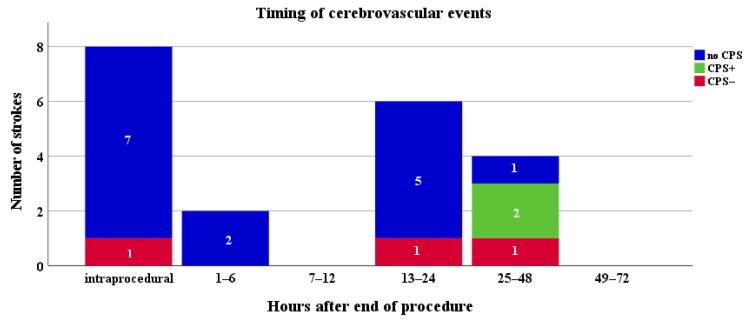
Timing of strokes in patients undergoing TAVR. NoCPS indicates TAVR without Sentinel™. Patients in whom both filters were correctly deployed are presented as CPS+, patients with incomplete/incorrect CPS deployment are indicated as CPS−.

**Figure 4 jpm-12-00320-f004:**
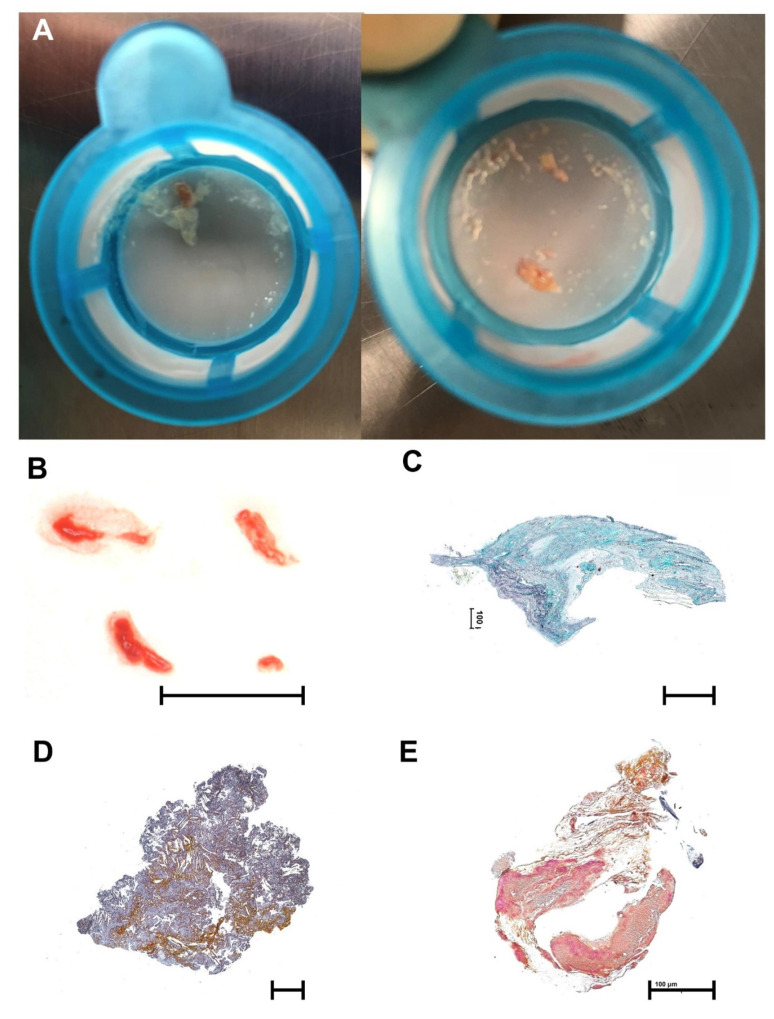
Debris captured in Sentinel™ filters. (**A**) Macroscopic images of fresh debris as collected immediately after the procedure; (**B**) Macroscopic image of fresh debris, black scale bar indicates 5 mm; (**C**–**E**) Tissue stained with trichrome, black scale bar indicates 400 µm; (**C**) valve tissue; (**D**) atherosclerotic plaque; (**E**) thrombus and fibrous tissue.

**Table 1 jpm-12-00320-t001:** Baseline characteristics.

	All *n* = 411	noCPS *n* = 198 (48.2%)	CPS *n* = 213 (51.8%)	CPS+ *n* = 189 (46.0%)	CPS− *n* = 24 (5.8%)	*p*-Value *
Clinical and laboratory parameters						
Age, y	80.4 ± 6.7	80.4 ± 6.8	80.4 ± 6.7	80.3 ± 6.6	81.2 ± 7.3	1.000
Sex, female, %	195 (47.4)	100 (50.5)	95 (44.6)	88 (46.6)	7 (29.2)	0.231
BMI, kg/m^2^	27.2 ± 5.2	27.0 ± 5.3	27.3 ± 5.2	27.3 ± 5.1	27.5 ± 5.9	0.474
Arterial hypertension, %	364 (89.2)	175 (89.3)	189 (89.2)	169 (89.9)	20 (83.3)	0.965
Diabetes mellitus, %	136 (33.4)	65 (33.2)	71 (33.6)	66 (35.1)	5 (21.7)	0.917
Peripheral artery disease, %	46 (11.3)	19 (9.7)	27 (12.8)	23 (12.2)	4 (17.4)	0.323
Coronary artery disease, %	260 (63.7)	127 (64.8)	133 (62.7)	116 (61.7)	17 (70.8)	0.665
Carotid artery stenosis, %	55 (13.5)	23 (11.7)	32 (15.2)	29 (15.4)	3 (13.0)	0.665
Hyperlipidemia, %	297 (73.2)	136 (69.4)	161 (76.7)	140 (74.9)	21 (91.3)	0.098
Prior stroke, %	30 (7.4)	15 (7.7)	15 (7.1)	11 (5.9)	4 (17.4)	0.834
Atrial fibrillation, %	169 (41.5)	86 (43.9)	83 (39.3)	71 (37.8)	12 (52.2)	0.353
COPD, %	46 (11.3)	20 (10.2)	26 (12.3)	23 (12.2)	3 (12.5)	0.500
Prior cardiac surgery, %	58 (14.1)	27 (13.6)	31 (14.6)	24 (12.7)	7 (29.2)	0.789
NYHA functional class						0.187
I, %	29 (7.6)	13 (7.1)	16 (8.0)	16 (9.0)	0.0	
II, %	113 (29.4)	50 (26.9)	63 (31.7)	55 (31.1)	8 (36.4)	
III, %	225 (58.4)	115 (61.8)	110 (55.3)	98 (55.4)	12 (54.5)	
IV, %	18 (4.7)	8 (4.3)	10 (5.0)	8 (4.5)	2 (9.1)	
CCS ≥ III, %	36 (8.9)	18 (9.1)	18 (8.6)	15 (8.1)	3 (12.5)	0.853
Syncope, %	65 (16.8)	28 (14.9)	37 (18.6)	34 (19.2)	3 (13.6)	0.331
NT-proBNP, pg/mL	3844 ± 6517	4139 ± 6670	3584 ± 6413	3414 ± 6062	4898 ± 8721	0.406
Creatinine, mg/dL	1.34 **±** 0.91	1.35 ± 0.82	1.34 ± 1.00	1.34 ± 1.03	1.33 ± 0.63	0.869
Oral anticoagulation, %	139 (34.1)	74 (37.4)	65 (31.0)	55 (29.6)	10 (41.7)	0.171
SAPT, %	136 (33.1)	56 (31.5)	64 (34.9)	56 (34.4)	7 (38.9)	0.504
DAPT, %	101 (28.1)	45 (25.3)	56 (30.8)	52 (31.7)	4 (22.2)	0.247
EuroSCORE II, %	6.3 ± 5.9	6.4 ± 5.4	6.4 ± 6.4	6.2 ± 6.2	7.9 ± 7.6	0.997
Echocardiographic parameters						
AV mPG, mmHg	47 ± 15	46 ± 14	49 ± 17	50 ± 17	41 ± 14	0.035
AV pPG, mmHg	75 ± 23	73 ± 21	78 ± 25	80 ± 25	66 ± 20	0.030
AV Vmax, m/s	4.3 ± 0.7	4.2 ± 0.6	4.4 ± 0.7	4.4 ± 0.7	4.0 ± 0.6	0.111
AVA, cm^2^	0.7 ± 0.2	0.7 ± 0.2	0.7 ± 0.2	0.7 ± 0.2	0.7 ± 0.2	0.126
AS Stage, %						0.906
High gradient	279 (73.6)	134 (73.2)	145 (74.0)	131 (75.7)	14 (60.9)	
LFLG + LVEF < 50%	47 (12.4)	22 (12.0)	25 (12.8)	21 (12.1)	4 (17.4)	
LFLG + LVEF ≥ 50%	53 (14.0)	27 (14.8)	26 (13.3)	21 (12.1)	5 (21.7)	
AR ≥ II, %	28 (7.1)	16 (8.1)	12 (6.0)	11 (6.2)	1 (4.3)	0.409
MR ≥ II, %	67 (16.9)	39 (19.9)	28 (14.0)	22 (12.4)	6 (26.1)	0.118
TR ≥ II, %	69 (17.3)	38 (19.2)	31 (15.5)	26 (14.7)	5 (21.7)	0.331
LVEF, %	50 ± 9	49 ± 9	50 ± 9	50 ± 9	49 ± 10	0.315
Procedural characteristics						
Self-expanding valves, % †	263 (64.0)	134 (67.7)	129 (60.6)	116 (61.4)	13 (54.2)	0.133
Valve size, mm	27 ± 3	27 ± 3	26 ± 3	26 ± 3	27 ± 3	0.163
Conscious sedation, %	392 (95.4)	186 (93.9)	206 (96.7)	182 (96.3)	24 (100)	0.181
Procedure time, min	59 ± 30	57 ± 24	60 ± 35	60 ± 36	61 ± 27	0.301
Predilation, %	198 (49.0)	92 (47.2)	106 (50.7)	95 (51.4)	11 (45.8)	0.477
Postdilation, %	103 (25.6)	49 (25.3)	54 (25.8)	48 (25.9)	6 (25.0)	0.894
Fluoroscopy time, min	17 ± 9	16 ± 8	19 ± 9	19 ± 9	21 ± 9	< 0.001
Contrast, mL	144 ± 55	149 ± 59	139 ± 52	139 ± 51	138 ± 60	0.079
Implantation > 1 valve, %	7 (1.7)	4 (2.0)	3 (1.4)	2 (1.1)	1 (4.2)	0.716
Valve-in-valve, %	28 (6.9)	13 (6.6)	15 (7.1)	13 (6.9)	2 (8.3)	0.860

AR indicates aortic regurgitation; AS, aortic stenosis; AV, aortic valve; AVA, aortic valve area; BMI, Body Mass Index; CCS, Canadian Cardiovascular Society; COPD, chronic obstructive pulmonary disease; CPS, cerebral protection device; CPS+, Sentinel™ deployed correctly; CPS −, Sentinel™ deployed incorrectly/incompletely; DAPT, dual antiplatelet therapy; LFLG, low flow low gradient; LVEF, left ventricular ejection fraction; mPG, mean pressure gradient; MR, mitral regurgitation; noCPS, no Sentinel™ used; NT-proBNP, N-terminal pro brain natriuretic peptide; NYHA, New York Heart Association; pPG, peak pressure gradient; SAPT, single antiplatelet therapy; TR, tricuspid regurgitation; Vmax, maximal velocity. *, comparison between CPS and noCPS group. †, Self-expanding valves used: Portico™ (13.6%), Symetis Acurate Neo™ (34.0%), Medtronic Evolut R™ (14.6%), Centera Valve™ (1.0%), Allegra Valve NVT™ (0.7%).

**Table 2 jpm-12-00320-t002:** Outcome analysis.

	All *n* = 411	noCPS *n* = 198 (48.2%)	CPS *n* = 213 (51.8%)	CPS+ *n* = 189 (46.0%)	CPS− *n* = 24 (5.8%)	*p*-Value *
Cerebrovascular event at 72 h	20 (4.9)	15 (7.6)	5 (2.3)	2 (1.1)	3 (12.5)	0.014
Disabling stroke (%)	10 (2.4)	7 (3.5)	3 (1.4)	1 (0.5)	2 (8.3)	0.162
Non-disabling stroke (%)	8 (1.9)	6 (3.0)	2 (0.9)	1 (0.5)	1 (4.2)	0.162
TIA (%)	2 (0.5)	2 (1.0)	0	0	0	0.231
All-cause mortality at 72 h (%)	2 (0.5)	1 (0.5)	1 (0.5)	1 (0.5)	0	0.939
All-cause mortality at 12 months	51 (12.4)	32 (16.2)	19 (8.9)	15 (7.9)	4 (16.7)	0.026
Hospital stay (days)	7.5 ± 8.0	8.4 ± 9.6	6.7 ± 6.1	6.4 ± 5.4	8.6 ± 9.9	0.031

CPS indicates cerebral protection device; CPS+, Sentinel™ deployed correctly; CPS−, Sentinel™ deployed incorrectly/incompletely; noCPS, no Sentinel™ used; TIA, transitory ischemic attack.*, comparison between CPS and noCPS group.

**Table 3 jpm-12-00320-t003:** Logistic regression analysis - Cerebrovascular events at 72 h.

Univariate Regression			Multivariate Regression	
	Odds Ratio (95%CI)	*p*-Value	Odds Ratio (95%CI)	*p*-Value
Baseline characteristics				
Sex, female	0.727 (0.291–1.818)	0.496		
Age	1.007 (0.942–1.078)	0.829		
Body mass index	1.002 (0.916–1.096)	0.969		
Arterial hypertension	0.664 (0.187–2.362)	0.527		
Diabetes mellitus	1.365 (0.544–3.421)	0.507		
Peripheral artery disease	0.404 (0.053–3.087)	0.382		
Coronary artery disease	1.768 (0.630–4.967)	0.279		
Carotid stenosis > 70%	2.267 (0.789–6.508)	0.128		
Hyperlipidemia	1.498 (0.490–4.583)	0.479		
Previous stroke	-			
Atrial fibrillation	0.938 (0.375–2.347)	0.892		
COPD	1.369 (0.386–4.854)	0.627		
Previous cardiac surgery	1.574 (0.507–4.886)	0.432		
Previous syncope	1.538 (0.486–4.875)	0.464		
NYHA functional class	0.938 (0.515–1.708)	0.835		
CCS ≥ III	1.157 (0.257–5.198)	0.849		
NT-proBNP (log.)	0.680 (0.297–1.560)	0.363		
Creatinine	0.773 (0.375–1.592)	0.484		
Oral anticoagulation	1.618 (0.654–4.001)	0.296		
SAPT	1.898 (0.750–4.805)	0.176		
DAPT	0.676 (0.219–2.089)	0.497		
EuroSCORE II	1.019 (0.955–1.088)	0.566		
Echocardiographic parameters				
AV mPG	0.979 (0.946–1.014)	0.235		
AV pPG	0.993 (0.970–1.015)	0.519		
AVA	2.167 (0.165–28.388)	0.556		
AV Vmax	0.772 (0.373–1.598)	0.486		
LVEF	0.987 (0.942–1.035)	0.592		
Procedural characteristics				
Self-expanding valve	1.390 (0.563–3.433)	0.476		
Sentinel^TM^	0.293 (0.105–0.823)	0.020	0.239 (0.075–0.762)	0.016
Valve size	1.077 (0.927–1.250)	0.331		
Predilation	0.746 (0.294–1.896)	0.539		
Postdilation	1.042 (0.366–2.968)	0.938		
Procedure time	1.010 (1.001–1.020)	0.032	1.009 (0.969–1.075)	0.129
Implantation >1 valve	32.083 (6.620–155.493)	<0.001	16.710 (2.687–103.923)	0.003
Fluoroscopy time	1.051 (1.010–1.093)	0.014	1.021 (0.969–1.075)	0.436

AV indicates aortic valve; AVA, aortic valve area; CCS, Canadian Cardiovascular Society; COPD, chronic obstructive pulmonary disease; DAPT, dual antiplatelet therapy; log, logarithmized; LVEF, left ventricular ejection fraction; mPG, mean pressure gradient; NYHA, New York Heart Association; pPG, peak pressure gradient; NT-proBNP, N-terminal pro brain natriuretic peptide; SAPT, single antiplatelet therapy; Vmax, maximal velocity.

**Table 4 jpm-12-00320-t004:** Logistic regression analysis—Twelve-months mortality.

Univariate Regression			Multivariate Regression	
	Odds Ratio (95%CI)	*p*-Value	Odds Ratio (95%CI)	*p*-Value
Baseline characteristics				
Sex, female	1.712 (0.964–3.040)	0.067		
Age	0.999 (0.959–1.040)	0.956		
Body mass index	0.988 (0.930–1.049)	0.684		
Arterial hypertension	0.532 (0.259–1.093)	0.086		
Diabetes mellitus	1.563 (0.898–2.720)	0.114		
Peripheral artery disease	2.286 (1.173–4.457)	0.015	0.935 (0.319–2.744)	0.903
Coronary artery disease	1.530 (0.827–2.830)	0.175		
Carotid stenosis > 70%	1.653 (0.828–3.300)	0.154		
Hyperlipidemia	0.776 (0.429–1.401)	0.400		
Previous stroke	0.772 (0.240–2.479)	0.664		
Atrial fibrillation	2.335 (1.331–4.097)	0.003	1.343 (0.662–2.724)	0.413
COPD	1.066 (0.455–2.499)	0.883		
Previous cardiac surgery	1.926 (1.088–3.678)	0.047	0.611 (0.185–2.016)	0.419
Previous syncope	1.159 (0.562–2.393)	0.689		
NYHA functional class	1.882 (1.184–2.993)	0.008	1.212 (0.727–2.021)	0.460
CCS ≥ III	0.635 (0.198–2.040)	0.446		
NT-proBNP (log.)	1.369 (1.098–1.708)	0.005	1.180 (0.885–1.574)	0.259
Creatinine	1.162 (0.933–1.446)	0.180		
Oral anticoagulation	1.583 (0.906–2.767)	0.107		
SAPT	1.184 (0.638–2.197)	0.593		
DAPT	0.882 (0.444–1.749)	0.719		
EuroSCORE II	1.055 (1.025–1.085)	<0.001	1.034 (0.984–1.088)	0.186
Echocardiographic parameters				
AV mPG *	0.978 (0.958–0.998)	0.032	0.986 (0.963–1.011)	0.271
AV pPG *	0.983 (0.970–0.997)	0.018		
AVA	1.690 (0.342–8.345)	0.519		
AV Vmax *	0.609 (0.397–0.936)	0.024		
LVEF	0.982 (0.954–1.010)	0.199		
Procedural characteristics				
Self-expanding valve	0.796 (0.441–1.439)	0.451		
Sentinel^TM^	0.478 (0.262–0.873)	0.016	0.454 (0.222–0.931)	0.031
Valve size	1.058 (0.962–1.164)	0.246		
Predilation	0.888 (0.509–1.548)	0.674		
Postdilation	0.831 (0.425–1.622)	0.586		
Procedure time	1.014 (1.008–1.020)	<0.001	1.016 (1.005–1.027)	0.005
Implantation > 1 valve	2.755 (0.670–11.330)	0.160		
Fluoroscopy time	1.026 (0.999–1.054)	0.059		

AV indicates aortic valve; AVA, aortic valve area; CCS, Canadian Cardiovascular Society; COPD, chronic obstructive pulmonary disease; DAPT, dual antiplatelet therapy; log, logarithmized; LVEF, left ventricular ejection fraction; mPG, mean pressure gradient; NYHA, New York Heart Association; pPG, peak pressure gradient; NT-proBNP, N-terminal pro brain natriuretic peptide; SAPT, single antiplatelet therapy; Vmax, maximal velocity. * only AV mPG was entered in the simultaneous multivariate regression.

**Table 5 jpm-12-00320-t005:** Linear regression analysis—Length of hospital stay.

Univariate Regression			Multivariate Regression	
	Regression Coefficient (95%CI)	*p*-Value	Regression Coefficient (95%CI)	*p*-Value
Baseline characteristics				
Sex, female	0.173 (−1.385, 1.731)	0.827		
Age	0.126 (0.911, 0.241)	0.031		
Body mass index	−0.076 (−0.249, 0.098)	0.390		
Arterial hypertension	−0.703 (−3.225, 1.819)	0.584		
Diabetes mellitus	0.558 (−1.105, 2.200)	0.510		
Peripheral artery disease	0.845 (−1.632, 3.322)	0.503		
Coronary artery disease	1.831 (0.213, 3.449)	0.027	1.332 (−0.327, 2.990)	0.115
Carotid stenosis > 70%	−0.490 (−2.785, 1.804)	0.675		
Hyperlipidemia	−0.493 (−2.267, 1.282)	0.585		
Previous stroke	0.000 (−3.003, 3.003)	1.000		
Atrial fibrillation	0.515 (−1.076, 2.106)	0.525		
COPD	−0.488 (−2.955, 1.979)	0.697		
Previous cardiac surgery	−0.239 (−2.474, 1.995)	0.833		
Previous syncope	2.227 (0.126, 4.328)	0.038	1.834 (−0.309, 3.976)	0.093
NYHA functional class	0.737 (−0.265, 1.739)	0.149		
CCS ≥ III	−0.324 (−3.087, 2.440)	0.818		
NT-proBNP (log.)	2.263 (0.976, 3.550)	<0.001	0.426 (−0.293, 1.144)	0.244
Creatinine	0.650 (−0.205, 1.505)	0.136		
Oral anticoagulation	0.057 (−1.584, 1.699)	0.945		
SAPT	0.585 (−1.161, 2.332)	0.510		
DAPT	−0.072 (−1.899, 1.755)	0.938		
EuroSCORE II	0.117 (−0.013, 0.247)	0.078		
Echocardiographic parameters				
AV mPG	−0.030 (−0.084, 0.024)	0.273		
AV pPG	−0.003 (−0.040, 0.034)	0.871		
AVA	−5.240 (−9.953, −0.527)	0.029	−4.355 (−8.985, 0.275)	0.065
AV Vmax	0.077 (−1.194, 1.348)	0.906		
LVEF	−0.101 (−0.192, −0.011)	0.029	−0.072 (−0.166, 0.023)	0.136
Procedural characteristics				
Self-expanding valve	1.480 (−0.134, 3.094)	0.072		
Sentinel^TM^	−1.972 (−3.520, −0.423)	0.013	−2.474 (−4.075, −0.874)	0.003
Valve size	−0.015 (−0.301, 0.270)	0.915		
Predilation	−0.864 (−2.441, 0.713)	0.282		
Postdilation	−1.064 (−2.875, 0.748)	0.249		
Procedure time	0.023 (−0.003, 0.048)	0.088		
Implantation > 1 valve	3.386 (−2.634, 9.405)	0.269		
Fluoroscopy time	0.127 (0.036, 0.219)	0.006	0.196 (0.097, 0.294)	<0.001

AV indicates aortic valve; AVA, aortic valve area; CCS, Canadian Cardiovascular Society; COPD, chronic obstructive pulmonary disease; DAPT, dual antiplatelet therapy; log, logarithmized; LVEF, left ventricular ejection fraction; mPG, mean pressure gradient; NYHA, New York Heart Association; pPG, peak pressure gradient; NT-proBNP, N-terminal pro brain natriuretic peptide; SAPT, single antiplatelet therapy; Vmax, maximal velocity.

## Data Availability

Data presented in this study are available upon request from the corresponding author.

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
