# Peer review of "Cerebral Protection in TAVR—Can We Do Without? A Real-World All-Comer Intention-to-Treat Study—Impact on Stroke Rate, Length of Hospital Stay, and Twelve-Month Mortality"

_jpm, 2022, doi:10.3390/jpm12020320_

Round 1
Reviewer 1 Report
Dear Authors,
Cerebral protection in cardiovascular procedures around the aorta is of utmost importance. It saddens me these filters are not used routinely.
I would like you to address a number of my questions.
1) "two out of four first operators systematically used the device whenever possible" - in what circumstances was it not possible? only those mentioned in the paragraph of device description or any other?
2) "Antiplatelet therapy, if present, was continued." - SAPT/DAPT? what drugs were used? in what regimens? were any benefits observed?
3) "The two filters cover more than 90% of the cerebral blood flow, excluding only the left vertebral artery's territory." - citation is needed or explain your calculations. There are many competitive cerebral collateral flow pathways to make this assumption
4) "Sentinel use was not attempted in the presence of severe left carotid artery stenosis, defined as stenosis severity > 70%, or if it was known that the right radial access was not possible." why was ulnar, brachial or axillary access not used in these cases?
5) "during these 72 hours, patients were evaluated clinically every 6 to 12 hours (9,20,21) by a board-certified cardiologist." - what exactly was evaluated if the comprehensive evaluation by a neurologist was needed? how did you determine neurological compromise? these can be subtle
6) "In 24 patients (5.8% of the total study cohort; 11.3% of CPS; CPS-) only the proximal filter could be positioned correctly in the brachiocephalic trunk while the distal filter remained closed in the ascending aorta." why?
7) "In the Sentinel group (CPS) five cerebrovascular events were recorded. Of these, two occurred in patients with correctly positioned CPS (CPS+; 2/189, 1.1%). Both comprised cerebral regions supplied by the left vertebral artery." what was the anatomy of left vertebral? what were the symptoms?
8) "Of 20 cerebrovascular events, 10 (50%) events occurred periprocedurally (within 12 hours of TAVR)." why this time frame? what is the rationale for recording a stroke in 12 h after the filter was removed, how is it related to the device? separate group should comprise only those events occurring during the procedure.
Author Response
Dear reviewer,
thank you very much for the valuable annotations. We adjusted the text accordingly - attached you can find the point-per-point response.
Best regards

Reviewer 2 Report
Thank you for permitting me to review this manuscript
Introduction , please provide a clear sentence for primary and secondary outcome ,
please explain the last sentence in introduction and the last phrase page
We hypothesised that the systematic use of Sentinel would effectively reduce both TAVR-related cerebrovascular events, LOS, and twelve-month mortality.
it appears to me that there is no systematic use of sentinel as there is two groups ?
Please provide a picture or photo of the sentinel if indeed central illustration does represent it please check it with an arrow .
The treatement attribution is not classical , please clearly state that there was no difference in the indications , if so please explain the difference between sentinel patients and non sentinel patients , please confirm that there was no confounding factor bias because of the operators
Please detail the LOS criteria.
Please provide the rate of known correct deployement of sentinel in the litterature.
Impact on daily prctice
If the difference in one year survival is not significant it should only be mentionned in the discussion section , (not in conclusion , nor in impact on clinical practice).
Author Response
Dear reviewer,
thank you for your valuable comments - we adjusted the text accordingly. Attached, you can find our point-per-point response.
Best regards

Round 2
Reviewer 1 Report
Thank you, I appreciate the changes you made and am satisfied with responses.
Reviewer 2 Report
The authors have signiicantly improved the manuscript